# Bone Healing Process of a Multiple Humeral Fracture in a *Caretta caretta*: Clinical, Surgical, Radiographic and Histomorphometric Assessments

**DOI:** 10.3390/ani13030376

**Published:** 2023-01-22

**Authors:** Carmela Valastro, Mariasevera Di Comite, Serena Paci, Delia Franchini, Stefano Ciccarelli, Antonio Di Bello

**Affiliations:** 1Department of Veterinary Medicine, University of Bari “Aldo Moro”, Strada Provinciale per Casamassima Km 3, 70010 Valenzano, Italy; 2Department of Translational Biomedicine and Neurosciences, University of Bari “Aldo Moro”, Piazza Giulio Cesare 11, 70124 Bari, Italy

**Keywords:** flipper fracture, bone repair, histological examination

## Abstract

**Simple Summary:**

Surgical treatment of sea turtle flipper fractures and the bones’ subsequent repair process are not well documented in the scientific literature. This is the first case report describing the repair process of a multiple humeral flipper fracture in a sea turtle, after surgical stabilization, through periodic radiographic follow-up from 0 to 24 months. Due to the accidental death of the turtle 3 months after its release, it was possible to describe the progress of the fracture-healing process by histomorphometric analysis and to histologically compare the structure of the newly formed bone with the normal bone structure of these animals. Based on our findings in this case study, we may be able to obtain a better understanding of how repair tissue in marine turtles works, as well as helping with decisions regarding surgery.

**Abstract:**

This study describes the surgical treatment of multiple humeral fractures in a *Caretta caretta* sea turtle referred by the ‘Centro Faunistico del Parco Regionale Bosco e Paludi di Rauccio’, in the area surrounding the city of Lecce, in southern Italy. Radiographs showed an evident detachment of the distal humeral epiphysis, compatible with a type II Salter-Harris fracture, as well as a transverse fracture of the diaphysis. After the surgical fracture reduction, radiographic follow-up was performed at 2, 4, 12, 16, and 24 months, showing a progressive healing and the formation of poorly mineralized callus tissue. Unfortunately, three months after his release at sea, the turtle was caught dead at a depth of 40 m. Histological and histomorphometric examinations of the surgically treated humerus were carried out on the corpse to collect further information about the bone tissue repair mechanisms in these animals.

## 1. Introduction

Among sea turtle species, *Caretta caretta* is the most common in the Mediterranean Sea [1]. During the migration through the Mediterranean Sea between April and October, they are often found on the Apulian shores due to beaching, ingested foreign bodies, such as fishhooks, or traumatic lesions resulting from collision with a hull or entanglement in fishing gear. The most common sea turtle injuries reported in the literature are head trauma, carapace fractures, and flipper lacerations [2,3,4]. Depending on the severity of flipper injury, amputation need not be carried out immediately, but can wait until nonviability is declared following conservative therapy [5]. If conservative therapy was not effective or trauma is severe, amputation is generally considered, given the remarkable ability of these animals to retain swimming and buoyancy [6,7]. Spontaneous repair of the fracture, although it may often lead to misalignment, generally offers an acceptable functional recovery. No substantial data are available in the literature on the treatment of limb fractures in sea turtles, and the bone tissue repair processes in these animals is still poorly investigated. Some information has been reported about the repair process in some terrestrial reptiles (lizards, frogs), showing very long repair times and the formation of a larger quantity of fibrous callus as compared to the bone repair processes that occur in mammals [8]. In reptiles it is considered important to stabilize the fracture to reduce the pain and limit further damage to the bones and muscles [9,10]. The present study describes the surgical treatment of multiple humeral fractures in a sea turtle and the radiographic follow-up, performed until the time of his release at sea. Due to the accidental death of the sea turtle, caught in a fishing net, it describes the histological and histomorphometry data on the fractured humerus obtained 28 months after the surgery and 3 months after the release at sea of the turtle.

## 2. Materials and Methods

A juvenile sea turtle was presented to the Sea Turtle Clinic (STC) of Veterinary Medicine Department of the University of Bari with an entanglement lesion affecting the right front flipper. On admission, the sea turtle was measured and underwent a complete physical examination; curved carapace length (CCL) from notch to tip ranged 41 cm, curved carapace width (CCW) was 37 cm, and the weight was 18 Kg. The turtle appeared responsive but weak and dehydrated. Clinical evaluation of the musculoskeletal system performed out of the water showed swelling of the right front flipper and evidence of pain on deep palpation of the respective brachial muscle. A reduction in the right front flipper’s range of motion was observed when the turtle was examined in the water. Radiographic assessment, in dorso-ventral (D-V) and Caudo-Cranial (C-C) projections, indicated detachment of the distal epiphysis of the humerus, compatible with a type II Salter-Harris fracture, together with a transverse diaphyseal fracture (Figure 1). The turtle underwent general anesthesia, and the craniodorsal access to the humeral diaphysis was performed. The epiphyseal fracture was reduced by closed surgery and fixed using two crossed 2.5 mm Kirschner pins, introduced backward from the distal stump and brought out through the hyperflexed humerus-radio-ulnar joint. The same pins used to fix the epiphyseal fracture were inserted into the proximal stump to fix the diaphyseal fracture. After checking the stability of the fractures reductions, the surgically sectioned soft tissues were reconstructed, and post-operative radiographs were performed. Follow-up was performed after the surgery at 2, 4, 12, 16, and 24 months, when the turtle was released at sea. Three months after its release at sea, the turtle was caught dead in a fishing net at a depth of 40 m. To determine the evolution of bone repair from turtle release to death, the right humerus was removed for histological and histomorphometry analyses. Immediately after explant, the humerus was fixed in 4% buffered paraformaldehyde, dehydrated in ethanol, and embedded in methylmetacrylate. Serial cross sections, 750 µm thick, were cut at both the diaphyseal and epiphyseal fracture gaps using a circular diamond-bladed saw (Gillings Hamco) and were ground to a thickness of 100 µm. Sections were placed on a specimen holder and microradiographed using a microradiograph (Constant 1-K, Ital Structures, Italy) at a prefixed distance from the X- ray generator of 9.5 cm. X-ray exposure was set up at 8 kV and 14 mA. Contact microradiographs were obtained on Kodak high-resolution film (SO 343, Eastman Kodak Co., Rochester, NY, USA), developed with Kodak HC-110, fixed in Kodak UNIFIX, washed in distilled water and then airdried at room temperature. Sections were subsequently stained with 1% toluidine blue (pH 3.7) for mineralized tissue. Four separate levels were selected, three belonging to the mid-diaphyseal fracture gap, and one level to the epiphyseal fracture gap: the first and the fourth levels corresponded to the proximal and distal part of the callus, respectively, and the second and third levels to the center of the fracture (Figure 2). As a control, the left humerus, which had never been fractured, was processed with the same procedure. On the stained sections, using a Nikon DS-5 camera connected to a stereomicroscope (SMZ800, Nikon Europe B.V., Amstelveen, The Netherlands) and a DS camera control unit, the callus extension was measured and the amount of new laid down bone, cartilage, and fibrous tissue, expressed as percentage ratio of the entire section, were evaluated. Histomorphometry was performed using Nis-Elements BR analysis software (Nikon Europe B.V.).

## 3. Results

### 3.1. Post-Operative Findings

The post-operative radiographic examination confirmed the correct reduction of the diaphyseal fracture, bone realignment, and the good fixation of the detached distal epiphyseal stump (Figure 3). Fifty days later, the X-ray follow-up showed persistent stability of the joint and bone sclerosis, near the diaphyseal fracture site. After 70 days, the turtle could swim and move around comfortably; radiographic examination showed a better filling of the diaphyseal fracture line and almost complete repair of the epiphysis (Figure 4). Four months after surgery, the animal was in good general condition and had recovered the right front flipper’s normal function. Radiography showed a complete repair of the epiphyseal fracture and remodeling of the normal bone architecture, while the diaphyseal fracture still lacked unifying bone callus tissue (Figure 5). After one year, the flipper showed complete clinical recovery without signs of pain, and radiologically the fracture site was fixed, showing signs of remodeling of the fracture stumps, areas of thickened fibrous tissue, but still no appreciable unifying bone callus (Figure 6a). The good positioning and tightness of the means of fixation were radiologically confirmed, but the surrounding bone reabsorption and decalcification processes were evident. Removal of the intramedullary pins was deemed necessary. Radiograms performed after removal of the pins showed the presence of calcified sites within the tissue interposed between the two bone stumps (Figure 6b). Two years after the procedure, radiographic follow-up demonstrated a better filling of the fracture line and further remodeling of the stump margins (Figure 7).

### 3.2. Histological Findings

After 28 months, the histological examination of the 1st level of the right anterior flipper, removed after the animal’s death, showed mostly woven bone surrounding a wide medullary cavity with a thick layer of interposed fibrous tissue. Compact necrotic bone fragments could be identified inside the medullary cavity, showing numerous erosion cavities and surfaces (Figure 8a). No direct contact was observed between this necrotic bone and the newly-laid-down vital bone. The intermediate two levels (2nd and 3rd levels) presented three distinct tissues: vascularized woven bone, cartilage, and fibrous tissue. The newly-laid-down bone was often situated close to cartilaginous islands. These areas of cartilage tissue were densely populated with hypertrophic chondrocytes surrounded by a deep violet-stained capsule. A few large immature vascular cavities were observed within the cartilage next to the newly-laid-down bone (Figure 8b,c). At the 4th level, the primary bone formed an entire circular structure, in which a rough stratification was already appreciable. There were many large primary osteons presenting a thin, rather radio-opaque wall. The wide vascular channels of primary periosteal bone exhibited a prevalently radial direction, whereas in the internal layer the vascular channels were arranged longitudinally. Near the endosteum, secondary voluminous osteons were visible, delimited by evident cementing lines (Figure 8d). In some peripheral areas the cartilage was visible as a sheath surrounding the periosteal bone (Figure 8e). Fibrous tissue was observed at all levels.

### 3.3. Morphometric Findings

The diameters of the callus tissues were significantly larger than those of the control humerus diaphysis. In all the sections woven bone tissue neoformation was evident, though to a different degree at the various levels examined. At the proximal level (1st level), a complete bone sheath of compact, spongy tissue was present surrounding a wide medullary cavity, within which fragments of preexisting necrotic bone over extensive surfaces and with erosion cavities were clearly visible. On average, at the intermediate levels (2nd,3rd level) 20% of the space was occupied by compact bone with wide vascular channels, sometimes opening toward the periosteum, and 32% by cartilage, present largely on the medial side. Inside the medullary cavity there were still slight residues of preexisting necrotic bone. Finally, at the distal level (4th level), there was a complete compact bone sheath, partly stratified due to periosteal growth on preexisting bone portions, accounting for about 56% of the entire section. Moreover, in the preexisting bone, numerous secondary osteons could be seen (Figure 8f). The cartilage, which was sometimes calcified, extended over 6% of the surface, in a fairly thin layer confined to peripheral areas of the medial side of the callus. In the control bone the compact diaphysis occupied 74% of the section (Figure 1).

### 3.4. The Epiphyseal Fracture

The histological and micro-radiographic examination of the epiphyseal fracture (5th level) showed complete fracture healing. Slight differences in shape and in microscopic structures were observed between the fractured and the healthy control humerus: the compact bone appeared slightly thickened (+ 31%) whereas the spongiosa appeared rarefied (−30%). Additionally, the fractured epiphysis showed a different outline, probably due to a reaction to the stainless-steel pins (Figure 9a,b). No cartilage or fibrous tissue were observed at this level (Figure 2).

## 4. Discussion

Sea turtles’ biology predisposes them to a variety of injuries, including shark bites and human-induced trauma, such as entanglement and propeller damage [2,3,4,11]. Sea turtles were shown to have an unexpected capability to heal given proper supportive care and accurate management [7]. Injuries involve mainly the front flippers of sea turtles, because of their greater length and exposure than the hind flippers [6]. To ensure that the turtle returns to the sea in a fully functional state, the surgeon’s primary objective should be preserving the limb. In case amputation needs to be considered, this can be done after the evaluation of postoperative outcome [11,12]. Decision making for the treatment of fractures is much more challenging in marine turtles than in mammals. Fragmentary information is available in literature on the treatment and repair mechanisms of long bones in Chelonian [6]. Despite the fact that this type of traumatic injury is often the cause of admission to sea turtle rehabilitation facilities, few case studies have been published on healing processes, which are fundamental to treatment outcomes. The principles of fracture repair in reptiles are like those described for domestic species and focus on rigid stabilization, correct anatomical alignment, and minimal disruption to the surrounding soft tissues [13]. The little information currently available in the literature on this topic, such as review articles and textbooks, which primarily focus on freshwater turtles and tortoises, prompted us to investigate bone tissue repair mechanisms in sea turtles. Many of the basic elements of tissue response and wound repair are shared among all chelonians, although the size, aquatic nature, and physiology of sea turtles result in some additional challenges. Infections and injuries involving bones also exhibit many of the same responses observed in other animals, including osteolysis and formation of reactive or woven bone. In addition, the bones of sea turtles, like other chelonians, readily undergo avascular necrosis and formation of bone sequestrum [14]. No substantial data are available in the literature on the treatment of long bone fractures in sea turtles. The selection of the fixation type must be related to the marine environment, as well as to the difficulty of keeping these animals in captivity for a long time. External coaptation methods are impractical in an aquatic environment [13,14]. External fixation could be difficult to adapt to the conformation and movement of the limbs of sea turtles, and it would require a period of immobilization that cannot be achieved as it can be with terrestrial reptiles [6]. In our patient, we decided to use Kirshnner pins as synthesis devices because they allowed us to reduce both fractures through a single surgical access. In fact, the fracture of the physis was reduced through a closed retrograde way, and the diaphyseal fracture through the unique surgical access performed on the cranial surface of the flipper. Unlike in mammals, in marine turtles there are no known radiographic elements that help to identify the repair times of fractures. Based on the authors’ experience, it is impossible to rely on any radiologically visible signs of a periosteal reaction, rounding of the fracture stump margins, progressive closure and remodeling of the stumps, and above all no radiological evidence of the formation of unifying bone callus. All these elements, together with the lack of clinically assessable functional deficits, led us to perform the present research to obtain more consistent data about the efficacy of surgical treatment for these fractures. Our histomorphometric data showed an elevated bone formation exclusively at the proximal and distal levels of the fracture gap, whereas the intermediate levels were characterized by a greater extension of cartilage. Moreover, bone deposition around a wide medullary cavity was observed at all the levels examined. The distal and proximal portions were completely healed whereas the intermediate levels showed only a partial ossification. The coexistence of bone, cartilage, and fibrous tissue suggests that both direct and indirect osteogenic processes take part in the repair processes, that in any case were shown to be very long. A substantial difference from bone repair processes in mammals was the constant presence of a wide medullary cavity. Furthermore, the long time required for bone formation compared to small animals, and the lack of data on sea turtles’ bone healing, mean that it is not possible to define the time that a fracture has occurred.

According to the results of this case study, the treatment of fractures in chelones is much more challenging than in mammals. The accidental capture of the same dead turtle allowed us to evaluate the fracture repair process over a long period and demonstrate the types of repairs that occur. Based on this evidence, it is impossible to deduce by the lack of bone tissue radiologically evident, if it is a recent fracture and date the trauma by radiographic exams. The long-term evaluation of the turtle’s swimming ability led us to conclude that there is a discrepancy between the short functional recovery time and the long fracture repair time. These considerations were supported by the high number of sea turtles admitted annually at the STC occasionally showing flipper fractures. All this preliminary consideration allows us to conclude that, although the internal fixation used did not have the same limitations as a surgical treatment with external fixation and was removed after an adequate time, no advantage is obtained from the surgical treatment of flipper fractures.

## 5. Conclusions

This case study provides a better overview of the bone healing process in marine turtles and may offer some guidance for a better management of flipper fractures. In the case of the diaphyseal fracture a lack of correlation among the clinical data, radiographic images, and histomorphometry data was demonstrated. On the contrary, the epiphyseal fracture showed a complete correlation among the three types of data collected. It has still to be ascertained whether the prolonged repair time of the diaphyseal fracture is attributable to factors inherent to this animal species, such as the slower basal metabolism or different biomechanical loads in the aquatic environment. The impossibility of establishing when the fracture occurred, the management complications associated with sea turtles’ aquatic environments, and the discrepancy between the flipper’s functional recovery and the long fracture repair time, led us to conclude that the surgical fixation of flipper fractures do not provide any benefit to animals.

## Data Availability

Not applicable.

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
