# Peer review of "Bone Healing Process of a Multiple Humeral Fracture in a Caretta caretta: Clinical, Surgical, Radiographic and Histomorphometric Assessments"

_animals, 2023, doi:10.3390/ani13030376_

Round 1

Reviewer 1 Report

Thank you for a well written and interesting paper on the very much needed issue of sea turtle flipper fractures and best options for adequate treatment! As a whole, I think the paper is very relevant, well explained, with clear results, discussion and conclusion. The english needs to be reviewed before publication, but only for minor issues.

Please find here a few small comments that might help improve the document as it is now, and that I am sure will be easily addressed by the authors:

Lines 39-40: maybe I am not understanding the statement correctly, but I cannot see how the references used refer to this statement? They seem very specific (treatment of 3 head injuries and residual vascularization as prognosis) to conclude from them what the most common sea turtle injuries reported are. Maybe look for another, more broad, paper/report about main stranding cases in the area, or in the Med?

Lines 54-55: I am curious to know if you think the turtle was already dead, and captured dead by the nets, or if it was doing great and was accidentally captured by the nets and died there. Might be important to know if the turtle was doing well with this repaired flipper?

Figures 4 and 5: they both mention the fracture at 4 months... not sure if I am missing something, or if months are wrong.

Graphic 1: diagrams are too small to see them well, and text cannot be read

Lines 306-307: again, not sure the references are the best to support the general statement done here. Maybe a report on main causes for admission at rescue centres, or for strandings?

Lines 363-364: "...although the fixation used for surgical reduction was the best...". Why was it the best? Based on what criteria? I think this needs a small explanation.

Lines 364-365: for me personally, as a sea turtle clinical vet, the most importance sentence in the whole paper is this one: the fact that there is no advantage in surgically treating flipper fractures. But somehow it seems diluted among all the text, and this important conclusion is not mentioned in the final conclusions. It might be worth considering if it is possible to give this statement more "space" or "importance", maybe explaining it a bit further?

Reviewer 2 Report

General comments

The article presents an interesting case of surgical treatment and radiographical and histological evolution of a multiple humeral fractures in a Caretta caretta that, in spite of a suboptimal healing, clinically recovered the normal motility to the affected flipper so that the turtle was released at sea.

The particularity of this case study is represented by the opportunity to describe the fracture healing process in a subject affected by diaphyseal and physeal fracture of the right humerus surgically treated and periodically checked with radiographs, till 24 months after the surgery, and accidentally caught dead three months after its release. The most interesting aspect of the article is represented by the description of the fractures not only radiographically but also histologically comparing the normal bone tissue of the contralateral humerus to the fractured one.

My major concern is related to the definition of the result, since, in my opinion, the diaphyseal fracture resulted in a “viable nonunion” or, considering the species, in a “delayed union” (see Palmer R.H. Nonunion, delayed union and malunion. In: Mechanisms of Disease in Small Animals (3rdEd.) by Bojrab M.J. and Monnet E.). The evolution toward a “nonunion” or a “delayed union” is clearly visible in the radiographic controls in which it is also visible a bone lysis at the bone-pins interface, specially in the proximal stump, maybe consequent to slight instability to the fracture site. The function of the flipper clinically recovered, in spite of the “nonunion” or “delayed union”, and this is possible as stated also in the Discussion chapter by the Authors. It is also necessary to explain why the Authors decided to treat the diaphyseal fracture with intramedullary pins instead of plate and screws and this choice has to be discussed. 

I suggest making modifications to the text in light of my concerns.

Specific comments

-      Lines 2-3: Since it seams that the diaphyseal fracture did not completely repair or, however, it lead to a “nonunion” or a “delayed union”, I suggest to change the title in this way: “Multiple humeral fractures in a Caretta carettasurgically treated: clinical, radiographic and histomorphometric assessments”. 

-      Line 69: please replace “Postero-anterior (P-A)” with “Caudo-Cranial (C-C)”.

-      Line 73: please replace “applied” with “performed”.

-      Lines 74-76: it is not clear how the pins were introduced, maybe they were introduced first in the proximal stump, starting from the fracture line in a backward way? Please, rephrase it. 

-      Line 77: what do you mean with “lacerated tissues” the surgically sectioned ones or a pre-existent wound?

-      Line 78: please replace “X-rays” with “radiographs”.

-      Lines 90-91: you describe a “callus” and after a “fracture”, while, maybe, it is better to describe it as a “viable nonunion” of the diaphyseal fracture (the last radiograph shows a slight angle of the distal stump axis compared to the proximal one).

-      Line 191: please replace “Craniocaudal” with “Caudocranial”.

-      Lines 205-206: the radiograph shows a slight angle between proximal and distal stumps axes: was there a slight mobility to the fracture site?

-      Lines 228-246 Figure 8 legend: all the images need a reference measure.

-      Line 341: maybe the term “consistent” is better than “scientific”.

-      Lines 346-347: I suggest to delete this phrase since it is not correct consider repaired the fracture observing a single level.

-      Lines 351-352: I suggest to delete this phrase or justify it with the absence of reference studies.

-      Line 356: please replace “longer” with “long”.

-      Lines 357-359: I suggest to delete this phrase; dating a fracture was never previously and adequately discussed.

-      Lines 363-365: The final phrases are controversial because, simultaneously, the authors state that the surgical treatment was the best but useless. Please, rephrase them in light of my concerns.

-      Lines 366-374: the Conclusions have to be modified according to the changes in the previous chapters.
